# Honest signaling in academic publishing

**Leonid Tiokhin**[1]*, **Karthik Panchanathan**[2], **Daniel Lakens**[1], **Simine Vazire**[3], **Thomas Morgan**[4,5], **Kevin Zollman**[6]

**1** Department of Industrial Engineering & Innovation Sciences, Human Technology Interaction Group, Eindhoven University of Technology, Eindhoven, The Netherlands, **2** Department of Anthropology, University of Missouri, Columbia, Missouri, United States of America, **3** Melbourne School of Psychological Sciences, University of Melbourne, Melbourne, Victoria, Australia, **4** School of Human Evolution and Social Change, Arizona State University, Tempe, Arizona, United States of America, **5** Institute of Human Origins, Arizona State University, Tempe, Arizona, United States of America, **6** Department of Philosophy, Carnegie Mellon University, Pittsburgh, Pennsylvania, United States of America

* leotiokhin@gmail.com

**Data Availability Statement:** No data was generated or analyzed for the current study. The minimal data set for this paper consists solely of mathematical equations, which can be found in the manuscript and supplementary materials.

## Abstract

Academic journals provide a key quality-control mechanism in science. Yet, information asymmetries and conflicts of interests incentivize scientists to deceive journals about the quality of their research. How can honesty be ensured, despite incentives for deception? Here, we address this question by applying the theory of honest signaling to the publication process. Our models demonstrate that several mechanisms can ensure honest journal submission, including differential benefits, differential costs, and costs to resubmitting rejected papers. Without submission costs, scientists benefit from submitting all papers to high-ranking journals, unless papers can only be submitted a limited number of times. Counterintuitively, our analysis implies that inefficiencies in academic publishing (e.g., arbitrary formatting requirements, long review times) can serve a function by disincentivizing scientists from submitting low-quality work to high-ranking journals. Our models provide simple, powerful tools for understanding how to promote honest paper submission in academic publishing.

## Introduction

Imagine a world in which scientists were completely honest when writing up and submitting their papers for publication. What might such a world look like? You might start with the observation that, like everything else, scientific research varies. Some research is well thought-through, conducted rigorously, and makes an important contribution to science. Other research is poorly thought-through, conducted hastily, and makes little contribution to science. You might imagine that scientists know quite a bit about the quality of their research and do everything that they can to honestly communicate this information to readers. High-quality research would be written-up as "amazing", "inspiring", and "robust", while low-quality research would be written-up as "weak", "derivative" and "inconclusive". Scientists would then submit high-quality research to high-ranking journals and low-quality research to lower-ranking ones. Imagine, no deception. In such a world, peer- and editorial-review of submissions

**Funding:** LT and DL were supported by the Netherlands Organization for Scientific Research (NWO) VIDI grant 452-17-01. KZ was supported by the National Science Foundation (NSF) grant SES 1254291. The funders had no role in any aspects of this study, the preparation of the manuscript, or the decision to publish.

**Competing interests:** We note that Simine Vazire is a member of the PLOS Board of Directors and that Leonid Tiokhin and Daniel Lakens are co-founders of Red Team Market. This does not alter our adherence to PLOS ONE policies on sharing data and materials.

would be straightforward, because quality would be readily apparent. Publication would be quick and painless, journals would reliably distinguish high- from low-quality research, and journal rank would provide a good proxy for research quality. Sure, you might feel like a dreamer. But you wouldn't be the only one [1–8].

Obviously, we don't live in such a world. After all, scientists are only human. We sometimes struggle to explain why our work is important or forget to write up critical aspects of our research protocols. Like all humans, we also seek out information that confirms our pre-existing beliefs, use biased reasoning strategies to arrive at pre-desired conclusions, and self-deceive to present ourselves in a better light [9–11]. Such factors can lead us to overestimate the quality of our work. And even if we were perfectly aware of our work's quality, we are incentivized to present it in an overly positive light [6]. One consequence is that we increasingly describe our research as "amazing", "inspiring", and "robust", despite the fact that such research is increasingly rare [12, 13]. We then submit this research to high-ranking journals, in part, because our careers benefit from publishing in prestigious and impactful outlets [14–17]. As a consequence, journal editors and peer reviewers must invest considerable time and effort to distinguish high- from low-quality research. While this process does help to filter work by its quality, it is slow and unreliable, and depends on reviewers' limited goodwill [7, 18–20]. As a result, low-quality research gets published in high-ranking journals [21–23]. An unfortunate property of this system is that journal rank provides a poor proxy for research quality [24]. And to top it off, we are well aware of these problems, and regularly complain that the publishing process is "sluggish and inconsistent" [25], "painful" [7], and "a game of getting into the right journals" [8].

What can be done to remedy this situation? This paper addresses one set of impediments to achieving an honest and effective publishing system: the information asymmetries and conflicts of interest that incentivize scientists to deceive journals about aspects of their research. Note that the term "deception" is used to refer to behaviors in which individuals communicate information that does not accurately (i.e., honestly) reflect some underlying state. This paper makes no assumptions about the motives of individual scientists or the mechanisms by which deception occurs.

## Information asymmetries and conflicts of interest in academic publishing

Academic journals are vulnerable to deception. This vulnerability exists for two interrelated reasons. First, there are information asymmetries between scientists and journals, as scientists have more information about aspects of their research than is presented in a paper (e.g., what the raw data look like, the scientists' original predictions versus those reported, what occurred during data-collection and analysis versus what was written up) [26]. Second, there are conflicts of interest between scientists and journals. For example, scientists have an incentive to publish each paper in a high-ranking journal, but high-ranking journals prefer to publish only a subset of papers (e.g., those with rigorous methods, compelling evidence, or novel results). By getting research published in high-ranking journals regardless of its true value, scientists can reap the benefits of high-ranking publications without doing high-value research [17].

One dimension along which journals are vulnerable to deception is research quality, and such deception imposes costs on the scientific community. First, if low-quality research is "deceptively" submitted to high-ranking journals, editors and reviewers must waste time evaluating and filtering out low-quality submissions. This extra time burden reduces the efficiency of science. Second, because peer-review is imperfect, some low-quality papers will "slip through the cracks" and be published in high-ranking journals. As a consequence, any correlation between journal rank and paper quality will be reduced. This reduced correlation impedes

accurate decision-making, as scientists rely on journal rank to decide which papers to read, which research paradigms to emulate, and which scientists to hire and fund [3, 16, 17, 27]. Third, if low-quality research requires less time to conduct than high-quality research but can still be published in high-ranking journals, then scientists have little incentive to invest in high-quality research. This can result in adverse selection: high-quality research is driven out of the academic market until low-quality research is all that remains [26, 28].

The problem of deception in communication systems is not unique to academic publishing —whenever there are information asymmetries and conflicts of interest, there are incentives to deceive. Consider three examples.

1. A mother bird brings food back to her nest and must decide which nestling to feed. The mother prefers to feed her hungriest child, and thus benefits from knowing how much food each child needs. But each child may prefer to receive the food for itself. That is, the mother would benefit if her children honestly communicated their level of hunger, but each child may benefit from deceiving its mother by claiming to be the hungriest.

2. A family needs a new car and decides to buy from a used-car dealership. The family prefers to buy a car without mechanical defects, worn-out parts, or a history of major collisions. But the used-car salesman prefers to make as much money as he can. This means selling even unreliable cars. The family would benefit if the used-car salesman honestly communicated which cars were reliable and which weren't, but the used-car salesman may benefit from deceiving the family by claiming that an unreliable car is of high-quality.

3. A university department is hiring a new faculty member and invites three candidates to give job talks. All else equal, the department prefers to hire a rigorous scholar—one who carefully thinks through each project, uses rigorous methods, and transparently reports all results and analyses. But each candidate prefers to get a job, even if they are not rigorous. That is, departments would benefit if job candidates gave talks that honestly communicated their scholarly rigor, but candidates benefit from deceiving departments by only communicating information that makes them seem rigorous (even if they are not).

How can honesty be ensured despite incentives for deception? The theory of honest signaling can shed light on this question. In economics [29, 30] and biology [31, 32], signaling theory represents an attempt to understand how honest communication can exist in situations where there appear to be incentives for dishonesty. Below, we apply the logic of signaling theory to the publication process using a set of formal theoretical models. This formal approach has several affordances, including making assumptions clear and explicit, ensuring that arguments are logically coherent, and providing insights into questions that are difficult to intuit (e.g., should publishing be made as efficient as possible so that papers can be rapidly submitted and evaluated by reviewers, or might removing publishing inefficiencies have unintended consequences?). However, our approach also has important limitations (see Discussion).

## A simple model of academic publishing

For simplicity, assume that a scientist produces two kinds of papers: high-quality and low-quality (see Discussion). In addition to other features, a high-quality paper might be one that thoroughly describes prior research, is methodologically rigorous, conducts appropriate statistical analyses and sensitivity checks, honestly reports all analyses and measures (e.g., no *p*-hacking or selective reporting of positive results), and clearly distinguishes between exploratory and confirmatory findings (see [5] for other factors that affect research quality). In contrast, a low-quality paper may have fewer or none of these qualities. In reality, scientists may

disagree about which methods or analyses are best, but there is often consensus that certain practices reduce research quality.

Conditional on paper quality, the scientist decides whether to submit to a high- or low-ranking journal (because submission is conditioned on paper type, the proportion of high- or low-quality papers is not relevant to our model). Publishing in a high-ranking journal results in payoff $B$, while publishing in a low-ranking journal results in payoff $b$, where $B > b$. These payoffs represent all the benefits that a scientist may receive from publication, including prestige, promotion, citations, or an increased probability of obtaining future funding.

Journals have imperfect information about the quality of submitted papers and thus probabilistically determine submission quality. High-ranking journals accept high- and low-quality papers with probabilities $P_h$ and $P_l$, respectively. We assume that high-ranking journals are more likely to accept high-quality than low-quality papers ($P_h > P_l$). The screening process that makes high-quality papers have a higher probability of acceptance could occur via various mechanisms (e.g., editor screening; peer-reviewer evaluation). If a paper is rejected from a high-ranking journal, the scientist resubmits the paper to a low-ranking journal. We assume that low-ranking journals accept all submitted papers. This assumption makes the model easier to understand without affecting its generality (because only the ratio of high- to low-ranking acceptance probabilities affects scientists' expected payoffs) and the model results are identical if low-ranking journals probabilistically accept submitted papers (see S1 File). Further, as scientists receive a fixed payoff for low-ranking submissions, this payoff can also be interpreted as the fixed payoff to a scientist who cuts their losses and chooses not to submit their work for publication.

In this model and subsequent modifications, we focus on situations in which scientists strategically submit papers to journals and journals adopt fixed behaviors. However, our qualitative results generalize to a model in which journals are also strategic actors (see S1 File and S1 Fig). The assumption of fixed journal behaviors is reasonable if scientists can adjust their behavior on shorter timescales than can academic journals. Further, it allows us to simply illustrate the same mechanisms that ensure honesty in true signaling games with two-sided strategic interaction (e.g., differential benefits and differential costs) [33].

Given this model, where should a scientist send high- and low-quality work in order to maximize their payoffs?

A high-quality paper is worth submitting to a high-ranking journal instead of a low-ranking journal when:

$$BP_h + b(1 - P_h) > b$$

$$P_h(B - b) > 0 \tag{1}$$

A low-quality paper is worth submitting to a high-ranking journal instead of a low-ranking journal when:

$$BP_l + b(1 - P_l) > b$$

$$P_l(B - b) > 0 \tag{2}$$

Both conditions are satisfied if 1) a scientist can publish in a high-ranking journal with non-zero probability and 2) a high-ranking publication is worth more than a low-ranking publication. In other words, in this model, scientists benefit from submitting all papers to high-ranking journals, regardless of paper quality.

This illustrates a key conflict of interest in academic publishing. Scientists are incentivized to signal that their work is high-quality (even when it is not), whereas journals prefer to know the true quality of the work. However, this conflict can be resolved by changing publishing incentives. Making journal submissions costly is one mechanism for doing so.

## Submission costs

Now assume that submitting a paper for publication is costly. Such costs could include any aspect of the submission process that requires time (e.g., writing a compelling cover letter, meeting stringent formatting requirements, waiting for a journal's decision) or money (e.g., submission fees), independent of paper quality. These costs can be conceptualized as either originating from the scientist (e.g., a signal) or as being enforced by the journal (e.g., a screening mechanism) [34]. Assume that scientists pay a cost, $C$, to submit a paper to a high-ranking journal and a cost, $c$, to submit a paper to a low-ranking journal, where $B > C$ and $b > c$. All scientists pay a cost once, but those whose papers are rejected from the high-ranking journal and are re-submitted to the low-ranking journal pay both costs. For mathematical simplicity, we analyze a case where $c = 0$, and do not include $c$ in our analyses. The qualitative results are identical in cases where low-ranking submissions have a non-zero cost (see S1 File).

With the addition of submission costs, how should a scientist now behave? A high-quality paper is worth submitting to a high-ranking journal instead of a low-ranking journal when:

$$BP_h + b(1 - P_h) - C > b$$

$$P_h(B - b) > C \tag{3}$$

A low-quality paper is worth submitting to a high-ranking journal instead of a low-ranking journal when:

$$BP_l + b(1 - P_l) - C > b$$

$$P_l(B - b) > C \tag{4}$$

With submission costs, scientists only submit to high-ranking journals when the benefits of doing so outweigh the costs.

**Separating high-quality from low-quality papers.** Eqs (3) and (4) define the conditions under which each kind of paper should be submitted to a high-ranking journal. If we seek to separate high-quality papers from low-quality papers based on the journal to which they are submitted, then the ideal case is where scientists submit high-quality papers to high-ranking journals and low-quality papers to low-ranking journals. Such a separating equilibrium [35] exists when:

$$P_h(B - b) - C > 0 > P_l(B - b) - C$$

$$P_h(B - b) > C > P_l(B - b) \tag{5}$$

Introducing submission costs creates a range of parameters in which honest submission is possible. The key insight is that imposing costs can promote honesty. This occurs because scientists who have low- and high-quality papers pay the same submission cost but have different expected benefits [33, 36] when submitting to high-ranking journals, as low-quality papers are less likely to be accepted.

Honesty is possible when the cost of high-ranking submission, $C$, is larger than the expected added benefit of submitting a low-quality paper to a high-ranking journal, $P_l (B–b)$, but smaller

than the expected added benefit of submitting a high-quality paper to a high-ranking journal, $P_h$ (B–b). As high-ranking publications become worth more than low-ranking publications (larger values of B–b), larger submission costs are required to ensure honest submission; otherwise, scientists will be tempted to submit all papers to high-ranking journals. However, if submission costs are too large, no separation exists, as no paper is worth submitting to a high-ranking journal. As journals become better able to differentiate high- from low-quality papers (larger values of $P_h$–$P_l$), the range of conditions under which honesty can exist becomes larger. Consider a case where high-ranking journals accept most high-quality papers and reject most low-quality ones. A scientist who submits low-quality papers to high-ranking journals pays a cost for each submission, but rarely obtains a high-ranking publication. As a result, dishonesty is not worthwhile unless the rare high-ranking publication is extremely valuable. In an extreme case where journals cannot tell the difference between high- and low-quality papers ($P_h = P_l$), honest submission cannot exist, because scientists lack any incentive to condition journal submission on paper quality.

In the above model, all scientists received the same benefit for high-ranking publications—differential benefits arose because of different acceptance probabilities for high- and low-quality papers. However, differential benefits can also exist if low-quality publications in high-ranking journals yield lower payoffs than high-quality publications and the same logic holds. Thus, differential benefits of this kind can also ensure honest submission (see S1 File). Such differential benefits are plausible. For example, publications in high-ranking journals are preferentially chosen for direct replication [21–23] and may be more heavily scrutinized for errors and fraud [24, 37], which increases the probability that low-quality papers in high-ranking journals are detected.

**Differential costs.** In the previous model, all papers were equally costly to submit for publication, and honesty was maintained because high- and low-quality papers produced different expected benefits for scientists. Another mechanism by which costs can ensure honesty is via differential costs. Differential costs exist if signal costs vary conditional on a signaler's type [33, 38]. In the context of our model, this would mean that submission costs differ depending on paper quality. Examples of such differential costs could include peer-review taking longer for low-quality papers [39] or it being more difficult to present low-quality work as compelling research (just as it may be harder to write compelling grant proposals for bad ideas than good ones [40]).

Assume that scientists pay *ex ante* submission costs $C_l$ and $C_h$ to submit low- and high-quality papers, respectively, to high-ranking journals, where low-quality papers have higher submission costs ($C_l > C_h$). With differential costs, a high-quality paper is worth submitting to a high-ranking journal instead of a low-ranking journal when:

$$P_h(B - b) > C_h \tag{6}$$

A low-quality paper is worth submitting to a high-ranking journal instead of a low-ranking journal when:

$$P_l(B - b) > C_l$$

Rewriting this inequality to express low- and high-ranking submission costs in the same units, substitute $C_l = kC_h$, where *k > 1*. The inequality thus becomes:

$$P_l(B - b) > kC_h \tag{7}$$

A separating equilibrium, such that scientists submit only high-quality papers to high-ranking

journals, exists when:

$$P_h(B - b) > C_h > \frac{P_l(B - b)}{k} \tag{8}$$

As in the differential-benefits only model (Eq 5), honest submission is more likely when journals can reliably differentiate between high- and low-quality papers. The range of conditions for honesty becomes larger as submitting low- versus high-quality papers to high-ranking journals becomes relatively costlier (large values of $k$). Differential costs promote honest submission (regardless of whether scientists receive differential benefits) by reducing the relative payoff of submitting low- versus high-quality papers to high-ranking journals.

## Costs for resubmitting rejected papers

In the previous models, we assumed that initial submissions were costly, but that there was no unique cost to resubmitting rejected papers to low-ranking journals. Below, we modify this assumption by making resubmission costly. Resubmission costs may be important if papers rejected from high-ranking journals have a lower expected value due to an increased probability of being "scooped" or take longer to publish elsewhere because scientists must make formatting modifications. Alternatively, if decision letters and reviews are openly available [41], rejected papers may lose value if other scientists lower their assessment of a paper's quality based on previous negative reviews.

Consider a modified version of the differential-benefits model (Eq 5) such that re-submitting papers rejected from high-ranking journals has a cost, $c_r$, where $c_r < b$. A high-quality paper is worth submitting to a high-ranking journal instead of a low-ranking journal when:

$$BP_h + (1 - P_h)(b - c_r) - C > b$$

$$P_h(B - b + c_r) > C + c_r \tag{9}$$

A low-quality paper is worth submitting to a high-ranking journal instead of a low-ranking journal when:

$$BP_l + (1 - P_l)(b - c_r) - C > b$$

$$P_l(B - b + c_r) > C + c_r \tag{10}$$

A separating equilibrium, such that scientists submit only high-quality papers to high-ranking journals, exists when:

$$P_h > \frac{C + c_r}{B - b + c_r} > P_l \tag{11}$$

Because low-quality papers are more likely to be rejected than high-quality papers, resubmission costs are disproportionately paid by scientists who submit low-quality papers to high-ranking journals. This can ensure honesty, even when initial submissions are cost free ($C = 0$).

## Limiting the number of submissions: A cost-free mechanism to ensure honesty

We have thus far assumed that scientists could resubmit rejected papers to low-ranking journals. This is how academic publishing tends to work: scientists can indefinitely resubmit a paper until it is accepted somewhere [14]. However, such a system allows authors to impose a

large burden on editors and reviewers. Might it be beneficial to limit submissions in some way? Below, we modify our model such that resubmissions are not possible. This could represent a situation in which papers can only be submitted a limited number of times [39] (see [4] for a related idea about limiting scientists' lifetime number of publications and [42] for a related proposal to limit scientists to one publication per year). Alternatively, it could represent a situation in which the cost of resubmitting papers to low-ranking journals is larger than the benefit of low-ranking publications ($c_r > b$), such that scientists have no incentive to resubmit rejected papers. Whether such reforms are feasible or desirable is unclear, but their logical consequences are important to understand.

For simplicity, assume that scientists can only submit each paper to one journal and that all submissions are cost-free. A high-quality paper is worth submitting to a high-ranking journal instead of a low-ranking journal when:

$$BP_h > b \tag{12}$$

A low-quality paper is worth submitting to a high-ranking journal instead of a low-ranking journal when:

$$BP_l > b \tag{13}$$

Thus, a separating equilibrium, such that scientists submit only high-quality papers to high-ranking journals, exists when:

$$BP_h > b > BP_l \tag{14}$$

Limiting the number of submissions can ensure honesty, even if submission is cost-free. As in the previous models, the range of conditions for honesty becomes larger when journals can reliably differentiate between high- and low-quality papers (large values of $P_h−P_l$). If high-ranking publications are worth much more than low-ranking ones ($B >> b$), honest submission can only exist if low-quality papers are usually rejected from high-ranking journals. In contrast, if high-ranking publications are not worth much more than low-ranking ones (small values of $B–b$), honest submission can only exist if high-quality papers are usually accepted by high-ranking journals.

Limiting the number of submissions works because scientists receive no payoff when a paper is rejected. In contrast, in the previous models, scientists could resubmit rejected papers to low-ranking journals and receive the smaller benefit, $b$. When the number of submissions is limited, scientists face an opportunity cost because submitting to one journal precludes submission to another. This disincentivizes deceptive submission as long as the expected value of a higher-probability, low-ranking publication outweighs the expected value of a lower-probability, high-ranking one. Note that, for illustrative purposes, we modeled an extreme case in which papers could only be submitted once. In the real world, less-strict submission limitations may be more feasible, but the mechanism by which limiting submissions ensures honesty would still apply.

## Relation to existing models in economics

Similar questions regarding signaling [30, 43, 44] and incentive structures in academic publishing [45–47] have a long history of study in economics. Most relevant to our paper, models have analyzed the optimal order in which authors should submit papers to academic journals, conditional on varying payoffs to publication, acceptance probabilities, publication delays, and authors' levels of impatience [39, 48–52]. For example, in one model of editorial delay and optimal submission strategies [48], authors know the quality of their papers and prefer to

publish each paper in a top journal, but top journals have higher rejection rates, paper submission is costly, and authors experience a time delay between rejection and resubmission. If submission costs and resubmission delays are sufficiently low, then authors' optimal policy is to initially submit papers to the very top journal and subsequently work down the journal hierarchy. However, as submission costs and resubmission delays increase, authors are incentivized to make initial submissions to lower-ranking journals, thereby decreasing journals' overall reviewing burden (see [49, 53] for the role of different types of publishing costs). In another game-theoretic model of the academic review process where authors have more information about paper quality than do journals, journals can increase the average quality of their published papers by increasing submission costs and reducing the noisiness of the review process, both of which disincentivize authors from submitting low-quality papers for publication [54]. Similar qualitative findings have emerged in several other models [50, 51].

Despite their potential to inform current discussions on scientific reform, the aforementioned models are not widely appreciated outside of economics. In part, this is due to their mathematical complexity, narrow target audience, and lack of connection to the burgeoning scientific-reform movement [5]. Our paper addresses these gaps by developing simple models that relate explicitly to proposals for scientific reform. We also go beyond economic theory and incorporate concepts from the theory of honest signaling in evolutionary biology (e.g., differential benefits and differential costs), which provide powerful conceptual tools for thinking about how to ensure honest communication. The explicit application of these models to recent proposals for scientific reform is essential, because the practical utility of models depends on the narrative within which they are embedded [55].

## Implications

Our models have implications for how to modify academic publishing to promote honest paper submission, and provide a range of insights regarding the repercussions of various publishing reforms.

## Publishing inefficiencies can serve a function

Scientists often complain about apparent inefficiencies in academic publishing, such as long review times, arbitrary formatting requirements, high financial costs to publication (e.g., Springer Nature's recent "guided open access" scheme charging €2,190 for editorial assessment and peer-review of manuscripts [56]) and seemingly-outdated norms (e.g., writing cover letters) [7, 8, 57–59]. As a result, academic journals lower submission costs by offering rapid turnaround times (e.g., *Nature*, *Science* [60, 61]), allowing authors to pay for expedited peer-review (e.g., *Scientific Reports* [62]), offering "short report" formats [63, 64], or recommending against writing cover letters [65]. Our models imply that such moves towards efficiency, even if well intentioned, may produce collateral damage because inefficiencies can serve a function: the costs associated with publishing reduce the incentive to submit low-quality research to high-ranking journals. Consider an extreme scenario in which high-ranking journals made submissions cost-free, removed all formatting requirements, and guaranteed reviews within 48 hours. If the benefits of high-ranking publications remained large, scientists would have even larger incentives to submit low-quality research to high-ranking journals, because the costs of doing so would be trivial.

Cutting costs could have additional repercussions. In signaling games where signal costs are too low to ensure separating equilibria, there exist "hybrid" equilibria where receivers of signals engage in random rejection behavior to prevent being exploited by deceptive signalers [33]. As a result, honest signalers are sometimes mistaken for dishonest signalers and honest

signals are rejected more frequently than in the higher-cost separating equilibrium. In academic publishing, well-meaning reforms to reduce publication costs could inadvertently lead to similar outcomes—scientists more frequently submit low-quality work to high-ranking journals, high-ranking journals counteract this by increasing rejection rates, and more high-quality papers are rejected as a consequence.

This illustrates why removing publishing inefficiencies without considering how they function in the larger scientific ecosystem may be counterproductive. An important question is whether the costs imposed on scientists are outweighed by the benefits of ensuring honest submission. This will not always be the case. For example, in research-funding competitions, the aggregate cost of writing proposals may outweigh the societal benefits of differentiating between high- and low-quality projects [40]. Similarly, the high signal costs necessary to ensure honest communication can leave both signalers and receivers worse off than in a system without any communication [66]. Making submission costs too large could also dissuade scientists from submitting to high-ranking journals (e.g., Springer Nature's. The fact that some high-ranking journals (e.g., *Nature*, *Science*, *PNAS*) continue to attract many papers and are preferentially targeted for initial submissions suggests that current submission costs are not this excessive [14, 67].

## Better peer review promotes honest journal submission

If journals preferentially accept high- versus low-quality research, given sufficient submission costs, scientists will not benefit from submitting low-quality papers to high-ranking journals. Ensuring this outcome requires that journals reliably differentiate between high- and low-quality work. In theory, peer review is the primary pre-publication mechanism for doing so. But in practice, peer-review regularly fails to differentiate between submissions of varying quality [18–20, 68]. Improving the quality of peer-review is a major challenge, and some minor reforms (e.g., short-term educational interventions) have had limited success [69]. This suggests that more substantial changes should be considered. The space of possibilities for such reforms is large, and includes outsourcing peer-review to highly-trained experts [70], harnessing the wisdom of crowds via various forms of open peer-review [71], allowing scientists to evaluate each other's peer-review quality [72], and supplementing peer review with "red teams" of independent critics [73]. However, because improving peer-review may be costly for journals, it is important to consider whether such costs are outweighed by the benefits of better discrimination between high- and low-quality papers [54].

## Transparent research practices promote honest journal submission

Improving peer reviewers' ability to distinguish low- from high-quality papers is difficult. In part, this is because reviewers lack relevant information to assess submission quality [26], a problem that is exacerbated by short-report article formats [64]. One solution is to reduce information asymmetries by mandating transparent and open research practices. Mechanisms for doing so include pre-registration, open sharing of data and materials [26], validating analyses before publication [41], removing word limits from the methods and results sections of manuscripts [74], requiring research disclosure statements along with submissions [75, 76], or requiring scientists to indicate whether their work adheres to various transparency standards (e.g., via the curatescience.org Web platform [77]).

It is worth noting that such reforms potentially increase the cost of peer-review, because reviewers spend extra time evaluating pre-registrations, checking raw data, and re-running analyses. Without compensating such costs (e.g., financial rewards), reviewers will have even fewer incentives to do a good job. A similar problem exists in animal communication: if

assessing signal veracity is too costly, receivers of signals may be better off by settling for signals that are less reliable [78]. This highlights the importance of ongoing efforts to reduce peer-review costs for papers with open data and pre-registered research (e.g., SMART pre-registration [79], machine-readable hypothesis tests [80]).

## Reducing the relative benefit of publishing low-quality papers in high-ranking journals promotes honest journal submission

Honest submission is more likely if low-quality, high-ranking publications are less beneficial than high-quality, high-ranking publications. Ways to generate such differential benefits include targeting high-ranking publications for direct replication [81, 82], or preferentially scrutinizing them for questionable research practices [75] and statistical/mathematical errors [83, 84]. This would increase the probability that low-quality papers are detected post-publication. Subsequently, scientists should pay costs that reduce the benefits associated with such publications. These might include financial penalties, fewer citations for the published paper, or reputational damage (e.g., fewer citations for future work or lower acceptance probabilities for future papers [85]). The fact that retraction rates correlate positively with journal impact factor suggests that high-ranking publications already receive extra scrutiny [24]. However, the fact that many high-ranking findings fail to replicate [21–23] suggests that current levels of scrutiny are not sufficient to ensure that only high-quality work is submitted to high-ranking journals [86].

## Costs for resubmitting rejected papers promote honest journal submission

As submission and resubmission costs become smaller, scientists have greater incentives to initially submit all papers to high-ranking journals, because getting rejected is not very costly. Resubmission costs solve this problem by making rejection costly, which disproportionately affects low-quality submissions. In principle, such costs could be implemented in several ways. If papers are associated with unique identifiers (e.g., DOI) and their submission histories are openly available, journals could refuse to evaluate papers that have been resubmitted too quickly. If journals preferentially reject low-quality papers, editors could wait before sending "reject" decisions, thereby creating disproportionate delays for low-quality submissions [39]. Note the ethical concerns regarding both of the above approaches (e.g., slowing down the communication of scientific findings). Another possibility is to make all peer-review and editorial decisions openly available, even for rejected papers, as is current policy at *Meta-Psychology* [41]. Although such a reform could introduce complications (e.g., generating information cascades or increasing the probability that authors are scooped pre-publication), it provides a plausible way to increase differential costs. For example, to the extent that low-quality papers are more likely to receive negative reviews, scientists will have fewer incentives to submit such papers to high-ranking journals, because receiving negative reviews could decrease the probability that the paper is published elsewhere, decrease its perceived scientific value once published, or harm scientists' reputations.

## Limiting the number of submissions (or rejections) per paper promotes honest journal submission

When scientists can indefinitely submit papers for publication and submission is cost-free or sufficiently low-cost, scientists are incentivized to submit all papers to high-ranking journals. Limiting the number of times that papers can be submitted or rejected solves this problem by introducing opportunity costs: submitting to one journal means losing out on the chance to

submit elsewhere [39]. If papers were assigned unique identifiers before initial submission, journals could potentially track submission histories and reject papers that had been submitted to or rejected from too many other journals. Such a policy would cause some papers to never be published. However, these papers could easily be archived on preprint servers and remain permanently available to the scientific community. Whether this would harm science as a whole remains an open question.

## Discussion

We have shown how honest signaling theory provides a tool for thinking about academic publishing. Making submission costly can disincentivize scientists from "deceptively" submitting low-quality work to high-ranking journals. Honest submission is more likely when 1) journals can reliably differentiate high- from low-quality papers, 2) high-quality, high-ranking publications are more beneficial than low-quality, high-ranking publications, and 3) low-quality papers are costlier to submit to high-ranking journals than high-quality papers. When journal submission is cost free or sufficiently low-cost, scientists are incentivized to submit all papers to high-ranking journals, unless 4) resubmission is costly or 5) the number of submissions is limited.

Our paper provides a formal framework for thinking about a wide range of deceptive publishing behaviors, without requiring any assumptions about scientists' motivations for engaging in practices that mislead readers about the quality of their work. That said, we provide just one formalization of the academic publishing process. In light of this, we note several potential extensions. We also discuss challenges associated with reforming the cost structure of academic publishing.

Just as experiments simplify reality to clearly establish cause-and-effect relationships, models necessarily simplify reality to serve as useful "thinking aids" for scientists [87–89]. As with all models, our analysis ignores many real-world details. For example, we do not address other factors that authors may consider when deciding where to submit a paper (e.g., whether a paper is a better fit for an interdisciplinary or specialty journal), although our model is generally relevant to cases where journals vary in selectivity and authors receive different payoffs for publishing in different journals. Further, we focus on papers varying in only quality, whereas papers and their perceived value actually vary along many dimensions (e.g., novelty, type of result (positive or negative), whether conclusions support scientists' preconceptions [6, 68, 90]). That said, our models are general enough to accommodate alternative interpretations of the single "quality" parameter. For example, we could have described papers as containing either positive or negative results, journals as preferentially accepting positive results, and authors as preferring to get all results published in high-ranking journals. If there were differential benefits to positive versus negative results, there would be some submission cost at which authors would only benefit from submitting positive results to high-ranking journals. It is worth noting that some emerging publishing formats, such as Registered Reports [91], ameliorate this issue by ensuring results-blind evaluation of submissions. More generally, future reforms would benefit from considering how publishing norms and incentives vary for different types of research and across different scientific fields.

Our models assume that papers vary in quality but do not address the process that generates different types of papers. A potential extension would be to allow scientists to influence paper quality by adjusting how much to invest in projects (e.g., sample size or methodological rigor [92, 93] as has been done in related models [45, 94–96]). Certain reforms (e.g., greater transparency, costs for rejected papers) decrease the payoff for low-quality research, and may incentivize scientists to produce more high-quality research in the first place. Further, although we

model one type of strategic interaction between authors and journals (see S1 File), this model is too simple to capture all interesting aspects of journals' strategic decisions (for other formalizations, see [49, 53]). For example, journals that invest more in peer review (e.g., soliciting more reviews) may be more likely to reject low-quality papers, but publish fewer papers than journals with lax peer-review policies. In turn, scientists might obtain larger benefits for publishing in more-rigorous journals. Depending on how journals are rewarded, this could be a viable strategy.

Future extensions could also analyze cases where authors have an imperfect estimate about a paper's quality [51, 54], either due to noise or strategic quality overestimation. Additionally, given that a work's perceived value is determined in part by social interactions between scientists [97], the stochasticity introduced by such processes might undermine honest submission (but see [98]). For example, if the cost of mistakenly submitting low-quality work to high-ranking journals is large, scientists may prefer to avoid such mistakes by only submitting to low-ranking journals.

We assume that submission costs vary only as a function of paper quality and journal type. However, in the real world, relative submission costs depend on other factors. For example, well-funded scientists with big labs can easily pay submission fees and offload costs onto junior lab members (e.g., writing grants or cover letters), whereas lone scientists with minimal funding are less capable of doing so. All else equal, this predicts that better-funded scientists will be more likely to submit low-quality work to high-ranking journals. Our models also assume that the benefits of high- and low-ranking publications are invariant across scientists. In the real world, the benefits of publication depend on other factors (e.g., career stage, scientific values). For example, well-established scientists may benefit less from high-ranking publications or, alternatively, may prefer to file-drawer a paper instead of submitting it to a lower-ranking journal.

It is also important to extend our models to allow for repeated interactions between scientists and journals. Several existing models of repeated signaling provide a starting point for doing so. Repeated interactions can ensure honesty if deceivers receive a bad reputation (e.g., are not believed in future interactions), thereby missing out on the benefits of long-term cooperative interactions. If deception is easily detected, receivers can simply not believe future signals from a partner who has been caught being deceptive [99]. If deception is detected probabilistically, receivers can more easily detect deceivers by pooling observations from multiple individuals to form a consensus [100]. And if deceptive signals are never detected but can be statistically detected in the long run, receivers can monitor the rate of signaling and forgo interactions with individuals who signal too frequently [101]. Similar reputation-based mechanisms can promote honesty in academic publishing. Journals that catch scientists engaging in misconduct can ban future submissions from those scientists. If editors and reviewers have information about the quality of authors' past publications, they can obtain a better estimate of a current submission's quality. Although such non-anonymous peer-review could introduce biases into the review process, complete anonymity would prevent editors and reviewers from basing publication decisions on scientists' history of producing low- or high-quality work.

Other extensions could incorporate the dynamics of author-journal interactions, as has been done in the signaling literature [102, 103]. This could be important, as dynamical models reveal the probability that populations reach different equilibria, as opposed to only establishing equilibrium stability. Real-world academic publishing does involve dynamic interactions between authors and journals–changes in journal policy in one time period affect optimal submission behavior in subsequent time periods, and journal editors who make policy changes may themselves be affected by such changes when they submit future papers [45].

Our models do not address whether submission costs harm or benefit science. In biological signaling theory, it is well-established that the high signal costs necessary for a separating

equilibrium can leave all parties worse off than in a system with lower signal costs and partial or complete pooling of signals across types [33, 66]. Given the difficulty of optimally calibrating submission costs, future work could extend our analysis to determine what combination of honesty and submission cost would lead to the most desirable scientific outcomes (see [53]). It is also worth noting that submission costs can exacerbate inequalities, as submissions may be costlier for individuals with fewer resources (e.g., early-career researchers, scientists in developing countries). One solution is to make submission costs conditional on scientists' ability to pay them [39]. This might mean faster review times for early-career researchers or lower submission fees for scientists who have limited funding.

Although we have focused on the utility of signaling theory for understanding reforms to academic publishing, existing theoretical frameworks from many disciplines will provide complementary insights. Some of these include economic theories of markets with asymmetric information [43] and public goods [85], cultural evolutionary theory [104] and its relevance to the scientific process [81, 94], and statistical decision theory [105]. Drawing on diverse theoretical frameworks will improve our ability to implement effective reforms and sharpen our intuitions about how incentives are likely to affect scientists' behavior. It will also improve our theoretical transparency, which has arguably lagged behind improvements in empirics [104, 106–108].

## Conclusion

How can we feasibly reform academic publishing to make it more honest, efficient, and reliable? We still lack definitive answers to this question. However, to the extent that we seek a publishing system in which journal rank correlates with paper quality, our models highlight several solutions. These include making submission costly, making rejection costly, making it costlier to submit low- versus high-quality papers to high-ranking journals, reducing the relative benefits of low- versus high-quality publications in high-ranking journals, improving the quality of peer review, increasing the transparency of submitted papers, openly sharing editorial decisions and peer-reviews for all submitted papers, and limiting the number of times that papers can be submitted for publication. Reforms based on these ideas should be subjected to rigorous theoretical and experimental test before implementation. Doing so will be our best hope for improving the efficiency and reliability of science, while minimizing the risk of collateral damage from the unintended consequences of otherwise well-intentioned reforms.

## Supporting information

**S1 Fig. Academic publishing with two-sided strategic interactions.** A decision tree with possible moves by both scientist and journals. In the first move, papers are randomly determined to be high- or low-quality. In the second move, the scientist chooses whether to submit the paper to either the high-ranking journal, with or without paying the submission cost, or to the low-ranking journal. The high-ranking submission cost is $C_h$ and $C_l$ for high-quality and low-quality papers, respectively. In the third move, a high-ranking journal decides whether to send the paper to harsh or soft peer review. Journals have imperfect information about paper quality. When high-ranking journals send papers to harsh peer-review, they accept high- and low-quality papers with probabilities $\bar{P}_h$ and $\bar{P}_l$, respectively. When high-ranking journals send papers to soft peer-review, they accept high- and low-quality papers with probabilities $\bar{\bar{P}}_h$ and $\bar{\bar{P}}_l$, respectively. Low-ranking journals accept all submissions. Papers rejected from high-ranking journals are re-submitted to low-ranking journals (not depicted). Dotted lines depict the journal's information sets. For each node in an information set, the journal does not know at which node they are.
(TIF)

**S1 File. Supplementary analyses and discussion.**
(DOCX)

# Acknowledgments

We thank Anne Scheel, Peder Isager, and Tim van der Zee for helpful discussions, and Carl Bergstrom for initially pointing us to the relevant economics literature. We thank Katherine Button, Barbara Spellman, Marcus Munafo and several anonymous reviewers for constructive feedback on previous versions of this paper.

# Author Contributions

**Conceptualization:** Leonid Tiokhin, Karthik Panchanathan, Daniel Lakens, Simine Vazire, Thomas Morgan, Kevin Zollman.

**Formal analysis:** Leonid Tiokhin, Karthik Panchanathan, Kevin Zollman.

**Methodology:** Leonid Tiokhin, Karthik Panchanathan, Thomas Morgan, Kevin Zollman.

**Supervision:** Daniel Lakens.

**Visualization:** Leonid Tiokhin.

**Writing – original draft:** Leonid Tiokhin.

**Writing – review & editing:** Leonid Tiokhin, Karthik Panchanathan, Daniel Lakens, Simine Vazire, Thomas Morgan, Kevin Zollman.

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
