## [Decision Letter · Decision Letter 0]

27 Oct 2020

PONE-D-20-19925

Honest signaling in academic publishing

PLOS ONE

Dear Dr. Tiokhin,

Thank you for submitting your manuscript to PLOS ONE. After careful consideration, we feel that it has merit but does not fully meet PLOS ONE’s publication criteria as it currently stands. Therefore, we invite you to submit a revised version of the manuscript that addresses the points raised during the review process.

As you can see, the paper received two divergent reviews, though both were fundamentally positive about the paper. I too am fundamentally positive about the paper, but some of R1's concerns about the assumptions of the model did occur to me too. However, I am not suggesting that you change your assumptions, merely that you acknowledge their limitations. Of course, you may be persuaded by R1's challenge: in which case, by all means change your assumptions and modify the model--or better yet, run multiple models with slightly different assumptions. I will probably not submit the next version of this paper for further review, but make the decision myself.

We look forward to receiving your revised manuscript.

Kind regards,

Jonathan Jong, PhD

Academic Editor

PLOS ONE

Journal Requirements:

"The authors have declared that no competing interests exist.".

We note Simine Vazire's PLOS Board of Directors membership.

Please confirm that this does not alter your adherence to all PLOS ONE policies on sharing data and materials, by including the following statement: "This does not alter our adherence to  PLOS ONE policies on sharing data and materials.” (as detailed online in our guide for authors http://journals.plos.org/plosone/s/competing-interests).  If there are restrictions on sharing of data and/or materials, please state these. Please note that we cannot proceed with consideration of your article until this information has been declared.ii) Please include your updated Competing Interests statement in your cover letter; we will change the online submission form on your behalf.

"LT and DL were supported by the Netherlands

Organization for Scientific Research (NWO) VIDI grant 452-17-01. KZ was supported by the

National Science Foundation (NSF) grant SES 1254291. The funders had no role in any aspects

of this study, the preparation of the manuscript, or the decision to publish."

Reviewers' comments:

Reviewer's Responses to Questions

**Comments to the Author**

1. Is the manuscript technically sound, and do the data support the conclusions?

Reviewer #1: Yes

Reviewer #2: Yes

2. Has the statistical analysis been performed appropriately and rigorously? 

Reviewer #1: N/A

Reviewer #2: N/A

3. Have the authors made all data underlying the findings in their manuscript fully available?

Reviewer #1: Yes

Reviewer #2: Yes

4. Is the manuscript presented in an intelligible fashion and written in standard English?

Reviewer #1: Yes

Reviewer #2: Yes

5. Review Comments to the Author

Reviewer #1: This paper discusses an issue in academic publishing – the information asymmetry between authors and editors – and considers how the logic of costly signaling theory can be applied to ensure that only high-quality papers are submitted to high quality journals. Their model considers two types of papers and two types of journals: high quality and low quality. It argues that in the absent of costs, authors will always be incentivized to submit all papers to high-quality journals, but that low-quality papers being reviewed by or published in high quality journals is a problem. The authors consider how costs to submit papers can change the decision calculus so that rational authors will only submit high-quality papers to high-quality journals.

I found the paper to be an interesting exercise, but I wasn’t particularly convinced by its arguments. The model is very simple and relies a number of assumptions that either rarely hold or, in the case of submission costs, *always* hold, raising questions about the conclusions one can draw. I found the discussion around the model to be cursory and at times naïve, with some occasionally problematic recommendations. That said, I thought the approach was interesting, and I found no major errors in the analysis. Connecting signaling theory to academia is a good idea, and I found there to be useful food for thought in this paper. It could also provide the foundation for more detailed modeling work on the subject. Given the broad appeal of the topic, it seems like PLOS ONE could provide a good home for the paper, pending some considerations of the comments below.

I first want to talk about the basic assumptions of the model. The model assumes that

1) Scientists should be honest

2) Scientists can accurately assess the quality of their research

3) Journal rank is a mark of quality, not specificity.

4) All research must be submitted as a paper.

The focus is on (1), and on mechanisms to enforce honesty. The others are implied and not explored, but I think there are some real issues there.

First, the extent to which researchers accurately assess the value of their work is highly questionable. It may also be adaptive to overestimate the value of one’s work – scholars that undervalue their work may be selected against in a competitive institutional system.

Second, the distinction between high- and low-quality journals seems very artificial. Certainly some journals are known to be crap (e.g. some predatory journals), and some are very prestigious, but there are also many journals that are excellent in terms of editorial curation and reputation but somewhat narrow in scope – what are often called technical or specialty journals. The dilemma many authors face is whether their paper has broad enough appeal to be submitted to a fancy interdisciplinary journal like Nature or a more solid but niche society journal.

Third, the authors write that in their ideal scenario, “Scientists would then submit high-quality research to high-ranking journals and low-quality research to lower-ranking ones” (lines 9-10). But why? Why publish low-quality research at all, especially if it’s readily identifiable as low quality? I kept thinking about this throughout the analysis, especially in terms of the cost tradeoffs. It seems like a crucial but missing component was the decision to not submit anything at all. This would carry no cost (except maybe the sunk cost of having done the work) and provide no benefit. This also speaks to something that is glossed over but important. ALL paper submissions incur some cost. It is not trivial to write up the paper and give up time to have the paper under review. The fact that most journals require papers to not be under submission elsewhere is a real tangible cost for all submissions.

Another assumption made is that “high-quality” journals are always incentivized to publish quality research. However, they are also incentivized to publish things that will get attention, and to minimize the risk of needing a retraction due to quality. This probably correlates with quality as the authors describe it, but not always – it would be good to address this, since the model strongly relies on this assumption and might run into problems if journal editors are also acting strategically.

The authors main conclusions are that high quality journals should impose (or continue to impose) costs for submission to dissuade trivial submission of low-quality papers. There are a few concerns about this. First, as noted, all submission is at least somewhat costly. Second, journals have desk rejection, which is a relatively low-cost way to weed out the more low-quality papers. And third, and most importantly, imposing high costs to submission has a potential social consequence, which is that it will tend to preferentially favor researchers who can afford to pay such costs (and so for whom the costs are effectively less). This favors senior researchers at wealthy, prestigious institutions doing normal paradigmatic science. This seems to me to be a major downside and needs to be addressed in the authors’ discussion.

_______OTHER COMMENTS_______

The section “Relation to existing models in economics” is useful but probably belongs much earlier in the paper, before the detailed model description. Also, an important but missing reference the authors might consider engaging with is:

Crawford and Sobel (1982) Strategic information transmission. Econometrica 50.

In the description of high vs. low quality research, they write (lines 112-116):

“A high-quality paper might be one that thoroughly describes prior research, is methodologically rigorous, conducts appropriate statistical analyses and sensitivity checks, honestly reports all analyses and measures (e.g., no p-hacking or selective reporting of positive results), and clearly distinguishes between exploratory and confirmatory findings (see (5) for other factors that affect research quality). In contrast, a low-quality paper may have fewer or none of these qualities.”

I would argue that you can have low quality research that has all these properties but fails to ask meaningful or important questions. This isn’t just about impact, it’s about insight. Indeed, many papers are likely rejected from “high quality journals” not because the research isn’t rigorous, but because they fail to be asking probing questions in a sufficiently deep fashion. Also, something to think about might be the fact that some research (often but not always “high quality”) may itself be more costly to conduct, and therefore authors may feel “entitled” to publishing in more prestigious journals to justify the investment, while they may feel more comfortable publishing “low-quality” research in third-tier journals because it doesn’t represent a substantial investment.

Lines 144-145: “papers are randomly determined to be high- or low-quality.”

In reality, there may be many more low quality than high quality papers. Would a skewed ratio change the calculus?

Lines 165: “This illustrates a key conflict of interest in academic publishing.”

The conflict here is that without costs, scientists are always incentivized to submit everything to “high-quality” journals, but HQ journals only want to publish HQ work. This being a conflict is contingent on a number of assumptions holding, including the absence of costs. But as noted above, there are almost always costs. There are costs to writing up results, for example, so another option is to just not submit. Consider this paper, which found that most null results are never even written up:

Franco et al. (2014) Publication bias in the social sciences: Unlocking the file drawer. Science 345.

This comment also impinges on the assumption to make the cost of submission to LQ journals c = 0, which they say “does not affect our model’s generality.” (line 179). This is only because the choice is between submit high or submit low. If another option is submit nowhere, which imposes no cost and causes no benefit, there might be cases in which a strategy could be “go big or go home” — submit to high-ranking journal, and if rejected, submit nowhere. This reduces the generalizability of c = 0.

In terms of deriving values of C to separate HQ and LQ papers, “The key insight is that imposing costs can promote honesty” (line 200). Maybe, but it also seems likely calibrating those costs will often be quite difficult. Especially since the costs and benefits are not constant for all individuals, but vary by research career stage, current prestige of position, workflow to submit and re-submit, etc. Further, if P_h is similar to P_l, there is a very narrow range of C, which interplays with the variation between individuals on both B and C.

Lines 207-209: “If high-ranking publications are worth much more than low-ranking publications (large values of B – b), large submission costs are required to ensure honest submission; otherwise, scientists are tempted to submit all papers to high-ranking journals.”

Not necessarily, if P_h is low. In that case, costs could still be low. Perhaps this suggests that HQ journals should just be more selective?

Regarding limiting the number of submissions, they cite suggestions to limit scientists’ lifetime number of publications or to limit scientists to one publication per year (lines 289-291). This example isn’t really appropriate, because it’s talking about limiting the number of publications, not the number of submissions.

In the Implications, the authors first reiterate the main conclusion: “the costs associated with publishing reduce the incentive to submit low-quality research to high-ranking journals” (lines 334-336).

They seem to have ignored desk rejection, which is the main weapon that journals have against wasting time with low-quality submissions. In truth, that might be enough. You haven’t talked at all about decision calculus for the journals. Why should they become more or less selective? What criteria should they use to select papers?

Lines 337-339: “If the benefits of high-ranking publications remained large, scientists would have even larger incentives to submit low-quality research to high-ranking journals, because the costs of doing so would be trivial.”

An interesting counterexample might be Sociological Science, which guarantees up/down decisions in 30 days and does not allow major revisions. Their rejection rate is high, and they have established a good reputation.

Lines 405-406: “When submissions and resubmissions are cost free…”

Again, I think this is almost never the case in practice. Especially if there’s the additional choice of not submitting at all.

Lines 409-411: “If journals preferentially reject low-quality papers, editors can wait some time before sending authors “reject” decisions, thereby causing disproportionate delays for low-quality submissions.”

Jesus Christ. No. Editors that do this should be punched in their stupid faces.

Lines 424-426: “Limiting the number of times that papers can be submitted or rejected…”

Beyond this being an impossible level of top-down control and overly paternalistic, such a limitation already occurs in practice because most journals have a policy that papers shouldn’t be submitted elsewhere.

Lines 472-473: “we note several extensions.”

Rather than simply listing extensions, which is about what you could have done but didn’t, why not spend some more time talking about the limitations and caveats of your conclusions based on your current model’s assumptions?

As a final note, something I kept asking myself while reading it was: Is this a high quality paper? Is PLOS ONE a high quality journal? Were the authors able to accurately assess the quality of their own research and use their model to help them decide where to submit? This is asked mostly rhetorically, but also seriously to the extent that it forces the question: are there real lessons to be drawn from this?

Reviewer #2: I really liked this paper. I thought it was clear, simple has a valuable insight, and useful prescriptions.

The basic idea is straightforward (and well explained): there is a fundamental informational and incentives problem in publishing—authors have an incentive to over-sell their work and have information about its value that the readers cannot as easily access, such as how much they had to twist the result or model or citations to make the results sound compelling and novel and interesting. The authors do a good job summarizing some of the inefficiencies this information+incentives problem creates. And then discuss how this problem can be understood and targeted using a standard costly signaling framework. Namely, in order to help readers differentiate between high and low quality papers, its essential that high quality papers can get into higher quality journals with lower relative cost—e.g. by having more attentive or better equipped referees, academic system that gives less credence to bad papers in good journals, or makes journal submission more onerous, misrepresenting results harder, or limits the frequency of submissions. I Think these are all nice insights and valuable prescriptions, and costly signaling proves to be a useful perspective to look at this problem.

Perhaps I missed something significant that other referees will catch, but on my reading, I couldn’t think of anything I would want changed in this paper or any reason to prevent its publication.

6. PLOS authors have the option to publish the peer review history of their article (what does this mean?). If published, this will include your full peer review and any attached files.

Reviewer #1: No

Reviewer #2: **Yes: **Moshe Hoffman

---

## [Author Response · Author response to Decision Letter 0]

11 Dec 2020

Response to reviewers has been uploaded along with the other materials.

---

## [Editor Report · Decision Letter 1]

25 Jan 2021

Honest signaling in academic publishing

PONE-D-20-19925R1

Dear Dr. Tiokhin,

We’re pleased to inform you that your manuscript has been judged scientifically suitable for publication and will be formally accepted for publication once it meets all outstanding technical requirements.

Kind regards,

Wing Suen

Academic Editor

PLOS ONE

Additional Editor Comments (optional):

The paper provides a nice application of signaling theory to academic publishing. The authors have adequately addressed the referees' comments at the earlier round. However, I have two simple suggestions to make before the paper can go into production.

1. Figure 1 of the paper is not particularly helpful. In the interest of brevity, I would suggest taking out the figure.

2. Referee 1 had some concerns about the suggestion that "editors could wait before sending authors 'reject' decisions, thereby causing disproportionate delays for low-quality submissions." (Lines 439-440, page 17). I share a similar concern. I could understand the underlying logic of making submissions differentially more costly to induce a separating equilibrium, but the suggestion borders on unethical editorial behavior. Clearly inducing a separating equilibrium may be desirable, but is not the only objective that can override all other concerns. I would suggest that the authors either take out the said recommendation, or clearly spell out the competing ethical issues that needs to be considered.
---

## [Editor Report · Acceptance letter]

15 Feb 2021

PONE-D-20-19925R1 

Honest signaling in academic publishing 

Dear Dr. Tiokhin:

I'm pleased to inform you that your manuscript has been deemed suitable for publication in PLOS ONE. Congratulations! Your manuscript is now with our production department. 

Kind regards, 

on behalf of

Professor Wing Suen 

Academic Editor

PLOS ONE